# Higher-order unimodal olfactory sensory preconditioning in *Drosophila*

**Juan Martinez-Cervantes[1†], Prachi Shah[1†], Anna Phan[2], Isaac Cervantes-Sandoval[1,3]***

[1]Department of Biology, Georgetown University, Washington, DC, United States; [2]Department of Biological Sciences and Neuroscience & Mental Health Institute, University of Alberta, Edmonton, Canada; [3]Interdisciplinary Program in Neuroscience, Georgetown University, Washington, DC, United States

**Abstract** Learning and memory storage is a complex process that has proven challenging to tackle. It is likely that, in nature, the instructive value of reinforcing experiences is acquired rather than innate. The association between seemingly neutral stimuli increases the gamut of possibilities to create meaningful associations and the predictive power of moment-by-moment experiences. Here, we report physiological and behavioral evidence of olfactory unimodal sensory preconditioning in fruit flies. We show that the presentation of a pair of odors (S1 and S2) before one of them (S1) is associated with electric shocks elicits a conditional response not only to the trained odor (S1) but to the odor previously paired with it (S2). This occurs even if the S2 odor was never presented in contiguity with the aversive stimulus. In addition, we show that inhibition of the small G protein *Rac1*, a known forgetting regulator, facilitates the association between S1/S2 odors. These results indicate that flies can infer value to olfactory stimuli based on the previous associative structure between odors, and that inhibition of *Rac1* lengthens the time window of the olfactory 'sensory buffer', allowing the establishment of associations between odors presented in sequence.

## Editor's evaluation

This paper shows that *Drosophila* can perform olfactory unimodal sensory preconditioning, an example of higher-order conditioning that may guide behaviour through inferred value. This is of conceptual significance for the brain, behavioural, and to some extent, the social sciences, because it shows that a conditioned response to a stimulus can occur even when the stimulus itself was never paired with punishment, for example.

**\*For correspondence:**
ic400@georgetown.edu

[†]These authors contributed equally to this work

**Competing interest:** The authors declare that no competing interests exist.

## Introduction

Learning and memory are fundamental for animal survival in dynamic and noisy environments and for the cognitive abilities of humans. A significant amount of what we know on the biological basis of learning and memory has come from studying classical conditioning, also known as Pavlovian conditioning (*Heisenberg et al., 1985*; *Romanski et al., 1993*; *Uwano et al., 1995*; *LeDoux, 2000*; *Davis and Whalen, 2001*; *Dubnau et al., 2001*; *McGuire et al., 2001*; *Maren and Quirk, 2004*; *Fanselow and Poulos, 2005*; *Ehrlich et al., 2009*; *Johansen et al., 2010*; *Duvarci and Pare, 2014*; *Herry and Johansen, 2014*). During classical conditioning, learning depends on the contiguity of conditioned and unconditioned stimuli. Studying classical conditioning has provided essential insights into the molecular, cellular, and circuit basis of how the brain transforms sensory information into memories and how it uses these memories to drive behavior. Nevertheless, in humans and other animals, the instructive value of naturally occurring reinforcing experiences is acquired rather than innately

instructive and does not necessarily dependent on mere contiguity. For example, the value of money and its capacity to function as a reinforcer is learned rather than innate. Learning theory has postulated the idea that learned behavioral control uses two types of information. The first results from habits, policies, or cached values (e.g., Pavlovian conditioning – model-free learning) (*Jones et al., 2012*). This kind of information produces a rapid, efficient behavioral response but does not consider changes in the value of the expected outcome. The second type of information relies on the knowledge of the associative structure of the environment to infer value (model-based learning). In other words, acquiring new knowledge depends on creating an associative structure of external events, which is then used to create additional new meaningful associations (*Daw et al., 2005*; *Jones et al., 2012*; *McDannald et al., 2012*; *Lucantonio et al., 2014*; *Sadacca et al., 2016*). Several different types of higher-order conditioning are examples of this type of information. In higher-order conditioning procedures, neutral stimuli acquire the property to elicit conditional responses even though they have never been in contiguity with a reinforcer. Sensory preconditioning is one example of higher-order conditioning (*Giurfa, 2013*; *Giurfa, 2015*; *Todd et al., 2016*). In sensory preconditioning, two initially neutral stimuli (S1 and S2) are repeatedly presented in contiguity or in sequence (preconditioning phase); later, one of the stimuli (S1) is paired with a reinforcer (conditioning phase). After this, S2 will elicit a conditioned response even though it was never paired with the reinforcer, indicating that the preconditioning phase created an association between S1 and S2. The response to the preconditioned stimulus (S2) is different from the response to the reinforcer-paired S1 in that it is not based on a mere association; instead, it must infer value by virtue of knowledge of the associative structure of the task (*Jones et al., 2012*). Sensory preconditioning has been shown previously in bees and flies (*Müller et al., 2000*; *Brembs and Heisenberg, 2001*; *Guo and Guo, 2005*). Using a flight simulator (*Guo and Guo, 2005*), trained flies, using visual or olfactory cues to avoid certain flight directions. They reported sensory preconditioning using S1–S2 cues of different modalities (olfactory and visual). Here, we provide evidence of olfactory unimodal sensory preconditioning in fruit flies. We show that the presentation of a pair of odors (S1 and S2) before one of them (S1) is associated with electric shocks elicits a conditioned response not only to the trained odor (S1) but also to the odor it was previously paired with (S2). This occurs even if the S2 odor was never presented in contiguity with the negative reinforcer. These data provide evidence for unimodal sensory preconditioning in *Drosophila*. These findings open the door to better understand how a simple brain achieves sensory preconditioning with limited neurons and synapses, providing meaningful insights into how a more complex mammalian brain may solve similar problems.

## Results and discussion

The *Drosophila* mushroom bodies (MB) are brain structures made up by the axons of the Kenyon cells (KC), and are known to play a central role in the encoding of olfactory memories (*Heisenberg et al., 1985*; *Dubnau and Tully, 2001*; *McGuire et al., 2001*). During associative memory acquisition, positive or negative values are assigned to odors by associated reward and punishment, respectively. This reinforcement is achieved by the coincident activation of a sparse number of KC by an odorant and the activation of dopaminergic neurons (DAN), which innervate the MB, by the unconditioned stimulus. Different DAN synapse onto discrete zones of the MB, resulting in 15 tile-like compartments of the MB. Each of these 15 tiled areas in turn synapse onto the dendrites of distinct corresponding mushroom bodies output neurons (MBONs). Activation of different MBONs are then known to alter behavioral approach or avoidance (*Aso et al., 2014*; *Owald and Waddell, 2015*; *Roselli et al., 2021*). The memory of an odor associated with an electric shock (classical conditioning) is known to be partially encoded as a decreased response specifically to the trained odor (CS+) in a particular MBON known as MBON-γ1pedc>α/β (*Hige et al., 2015*; *Perisse et al., 2016*; *Cervantes-Sandoval et al., 2020*). Conversely, odors that have not been associated with an electric shock do not normally demonstrate decreased responses in the MBON-γ1pedc>α/β neuron (*Hige et al., 2015*; *Perisse et al., 2016*; *Cervantes-Sandoval et al., 2020*). Thus, we were extremely surprised to report in a previous publication that flies expressing the dominant-negative form of *Rac1^N17* (*Rac1DN*) in adult KC using the temperature-dependent tool gal80^ts (TARGET) (*McGuire et al., 2003*), displayed a depression to a non-shock associated odor in MBON-γ1pedc>α/β neurons (*Cervantes-Sandoval et al., 2020*). We decided to investigate the nature of this intriguing non-shock associated neural depression.

It has been reported that animals can link together neutral sensory cues presented in succession or simultaneously; if then one of these cues is associated with reinforcement, animals can display a response to the non-associated pre-linked cue; this is known as sensory preconditioning (*Brogden, 1939*; *Jones et al., 2012*). As mentioned above, this type of behavior is categorized as model-based learning where a value is *inferred* from the associative structure of the environment (*Jones et al., 2012*). It was also reported that expressing $Rac1^{N17}$ in KC significantly enhances trace conditioning, in which an odor is associated with an electric shock presented many seconds after odor offset – suggesting that inhibition of *Rac1* lengthens an olfactory 'sensory buffer' that later converges with the punishment signal (*Shuai et al., 2011*). Therefore, we reasoned that sensory preconditioning and *Rac1* inhibition could explain the observed depression to the non-trained odor we previously reported (*Cervantes-Sandoval et al., 2020*) and replicated this finding in pilot experiments (*Figure 1—figure supplement 1*).

To test for the possibility that we were observing sensory preconditioning in MBON-γ1pedc>α/β neurons, we exposed animals to a single pairing of 5 s of 4-methylcyclohexanol (MCH) (S1) and 5 s of 3-octanol (OCT) (S2) odors, separated by an interstimulus interval (ISI) of 1 s, 30 s, or 5 min, before classical aversive olfactory conditioning occurred only to S1 by pairing MCH (S1) with an electric shock (20 s of odor with four 90 V, 1.25 s electric shocks). This is also known as forward conditioning. Post-training calcium responses to both MCH and OCT were then recorded 5 min after conditioning in MBON-γ1pedc>α/β neurons using the *R12G04-lexA* driver to express GCaMP6f (*Figure 1A, B*). These calcium responses in MBON-γ1pedc>α/β were compared with the odor responses in backwards conditioned animals which serve as our procedural control. Backwards conditioned animals receive all the same stimuli as forward conditioned animals, except that the electric shock occurs before the odor during what would be the 'conditioning phase', thus classical conditioning does not occur. Two naive odors, penthyl acetate (PA) and ethyl lactate (EL), were also given to the animals post-conditioning to ensure that the experimental methods did not result in generalized MBON-γ1pedc>α/β changes to all odors (*Figure 1A, B*).

Using the typical imaging protocol with a 25- to 30-s ISI between the preconditioned odors, we and others (*Hige et al., 2015*; *Cervantes-Sandoval et al., 2020*) have shown there is no evidence of sensory preconditioning in the MBON-γ1pedc>α/β of control animals. That is, the MBON-γ1pedc>α/β does not display a decreased response to the S2 odor (*Figure 1C*), but does demonstrate the expected near-complete depression in response to the conditioned S1 odor. Lengthening the ISI to 5 min has the same effect on control flies (*Figure 1C*). Interestingly however, when we shorten the sensory preconditioning interval to 1 s, the MBON-γ1pedc>α/β neurons in control flies exhibit a decreased calcium response to the S2 odor, and to the S1 odor that is expected (*Figure 1C*). Throughout these experiments, MBON-γ1pedc>α/β responses to naive odors remained unchanged, indicating the decrease in calcium response to the S2 odor is specific to the preconditioning phase, and not the result of a generalized depression to all odors (*Figure 1C*). This suggests that shortening the ISI window during sensory preconditioning allows control flies to form some association between the S1 and S2 odors, thus resulting in an inferred predictive value of the S2 odor after animals were aversively conditioned to the S1 odor. These results demonstrate that flies show, at least at the physiological level, sensory preconditioning.

In contrast to control animals, flies expressing dominant-negative $Rac1^{N17}$ in the KC had previously been found to exhibit a decreased calcium response to the S2 odor when using an ISI of 30 s (*Figure 1D*, *Figure 1—figure supplement 1B*). This was also true when the sensory preconditioning ISI was decreased to 1 s (*Figure 1D*). The data from our control flies led us to speculate that the timing of the S1/S2 ISI dictates whether an association between odors S1 and S2 (MCH and OCT) occurs during the sensory preconditioning stage. Secondly, from prior data on Rac1's role in trace conditioning and our sensory preconditioning data, we suspected that *Rac1* inhibition in the KC extends the time period that allows for the S1/S2 association to occur. If this was true, we predicted that extending the ISI should eliminate sensory preconditioning in *Rac1DN* flies. When the ISI was increased to 5 min, no sensory preconditioning was observed in the *Rac1DN* animals (*Figure 1D*). Again, naive odor responses in these flies were unchanged. This widening of the ISI window to allow for an S1/S2 association is dependent on the inhibition of Rac1, since flies with the same genotype but raised at 18°C to prevent the expression of $Rac1^{N17}$ (using the TARGET system) showed the normal conditioning-induced odor depression to the S1 but no change to S2 with a preconditioning ISI window of 30 s (*Cervantes-Sandoval et al., 2020*).

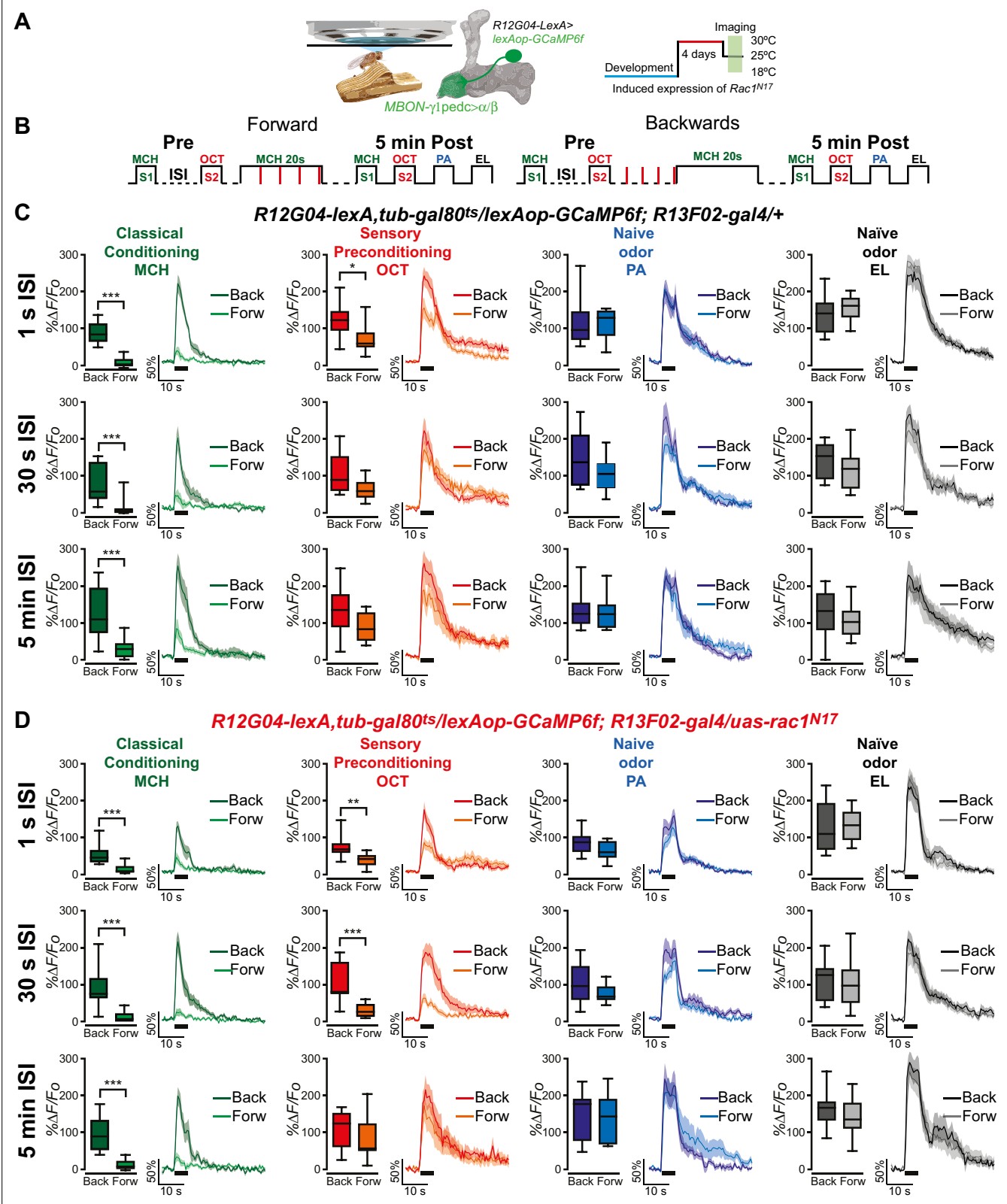

**Figure 1.** One-second interstimulus interval (ISI) during preconditioning induces depression to non-trained, pre-paired odor S2 in control flies. (**A**) Diagram of the in vivo, under the microscope training imaging setup, MBON-γ1pedc>α/β diagram, and $Rac1^{N17}$ expression induction. (**B**) Diagram of the experimental setup (odor schedule): preconditioning was induced by presentation of 4-methylcyclohexanol (MCH) and 3-octanol (OCT) (S1 and S2) with 1-s, 30-s, or 5-min ISI; then flies were aversively trained to MCH (S1) and 5 min later, post-conditioning responses were recorded. Responses

*Figure 1 continued on next page*

*Figure 1 continued*

were compared to flies trained using backwards conditioning. (**C**) S1 (MCH) responses were completely depressed 5 min after training in control flies preconditioned with 1-s, 30-s, or 5-min ISI. Non-parametric Mann–Whitney test p ≤ 0.0044; *n* = 8–10. A significant inhibition of post-trained response to S2 (OCT) was observed in control flies preconditioned with 1-s ISI. Non-parametric Mann–Whitney test p = 0.035; *n* = 10. In contrast, S2 (OCT) responses in control flies preconditioned with 30-s or 5-min ISI showed no significant change. Non-parametric Mann–Whitney test p ≥ 0.0630, *n* = 8–10. Neither naive odor (penthyl acetate [PA] and ethyl lactate [EL]) responses showed any significant reduction in flies preconditioned with 1-s, 30-s, or 5-min ISI. Non-parametric Mann–Whitney test p ≥ 0.0726, *n* = 8–10. (**D**) S1 (MCH) responses were completely depressed 5 min after training in flies expressing *Rac1^{N17}* preconditioned with 1-s, 30-s, or 5-min ISI. Non-parametric Mann–Whitney test p ≤ 0.0009; *n* = 8–12. A significant inhibition of post-trained response to S2 (OCT) was observed in flies preconditioned using a 1- or 30-s ISI. Non-parametric Mann–Whitney test p ≤ 0.0031; *n* = 8–12. In contrast, S2 (OCT) responses in flies preconditioned with 5-min ISI showed no significant change. Non-parametric Mann–Whitney test p = 0.4234, *n* = 8–9. Neither naive odor (PA and EL) responses showed any significant reduction in flies preconditioned with 1-s, 30-s, or 5-min ISI. Non-parametric Mann–Whitney test p ≥ 0.0726, *n* = 8–12. Boxplots represent distribution of %$\Delta F/F_o$ responses across the 5 s of odor presentation. The thick black bar below each trace represents the time of odor presentation. The asterisk's number shows N digits after the decimal.

The online version of this article includes the following source data and figure supplement(s) for figure 1:

**Source data 1.** One-second interstimulus interval (ISI) during preconditioning induces depression to non-trained, pre-paired odor S2 in control flies.

**Figure supplement 1.** *Rac1* inhibition induces depression to non-trained, pre-paired odor S2 in MBON-γ1pedc>α/β.

**Figure supplement 1—source data 1.** Rac1 inhibition induces depression to non-trained, pre-paired odor S2 in MBON-γ1pedc>α/β.

We tested whether this suspected sensory preconditioning-induced calcium depression might be the result of increased habituation of the non-associated S2 odor. We recorded olfactory responses in flies exposed to the same odor protocol except that the unconditioned stimulus was excluded. Results showed no depression to either S1 or S2 odor presented (*Figure 2—figure supplement 1A*). Going even further, we confirmed this observation by recording and evaluating changes in calcium responses after 10 repeated presentations of both MCH and OCT. Data were analyzed as previously described (*Hattori et al., 2017*). The ratio between the mean of three initial responses and mean of three last responses in MBON-γ1pedc>α/β was not significantly different between control and experimental genotypes (*Figure 2—figure supplement 1B*). Thus, the reduced calcium response we see in MBON-γ1pedc>α/β to the classically conditioned S1 odor and the sensory preconditioned S2 odor cannot be explained by a general habituation to the odors due to the experimental protocol.

*Rac1* has been implicated in memory forgetting, and it has been theorized that forgetting is fundamental for memory generalization (*Richards and Frankland, 2017*). In addition, (*Hige et al., 2015*) showed that a similar depression to the non-trained odor could result from a partial overlap in the neural representation of odors such as MCH and OCT. For these reasons, we tested whether the expression of *Rac1^{N17}* in KC may somehow increase that generalization between MCH and OCT, and that this may lead to the sensory preconditioned neural response that we are observing. For this, we trained animals using orthogonal odors, PA and EL, that are known to have little overlap in their KC representations. Surprisingly, similar results to MCH/OCT were obtained; we observed a depression to the non-trained S2 odor (EL) when *Rac1* function is inhibited in KC (*Figure 2*). Thus, this physiological response to sensory preconditioning can be observed for different odors and odor pairs.

To further challenge the assertion that the reduced calcium response in MBON-γ1pedc>α/β neurons to the S2 odor is a result of sensory preconditioning, we trained animals with forward or backwards conditioning but excluded the pre-training odor presentation of S1 and S2. When we eliminated the preconditioning association between the S1 and S2 odors, the results showed significant depression to the classically conditioned S1 odor that was associated with the shock, but no depression to any other odors (*Figure 3*). These results demonstrate that the depression to the non-trained S2 odor was not due to a general depression to odors or generalization of learned aversion. Instead, depression of the non-trained S2 odor is dependent on the animal learning the association between S1 and S2 during the preconditioning stage. These results further support the hypothesis that we are observing physiological sensory preconditioning.

We then tested if sensory preconditioning could be observed at the behavioral level. For this, we trained the flies using an Arduino microcontroller for precise odor delivery by controlling solenoids automatically. Using this Arduino system, we trained animals as follows (*Figure 4—figure supplement 1A*). Control flies were presented with a single pairing of the odors (5 s pulse each) with a 1-s ISI. Using the standard aversive olfactory conditioning paradigm, flies were then conditioned by the presentation of 1 min of S1 odor along with 12, 90 V shocks. Memory was tested right after training in

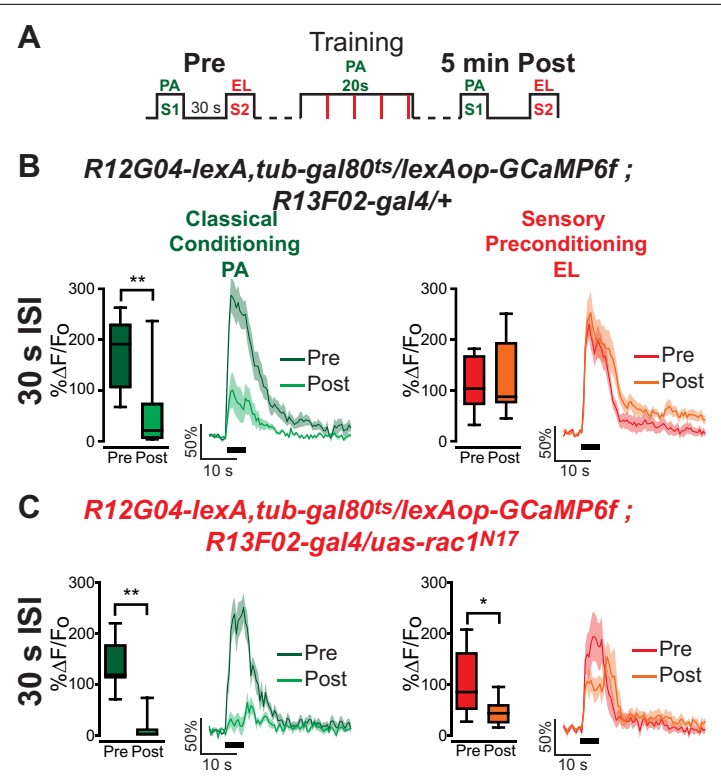

**Figure 2.** *Rac1* inhibition induces depression to sensory preconditioned odor S2 in MBON-γ1pedc>α/β using a different odor pair with orthogonal representation. (**A**) Diagram of experimental setup (odor schedule): preconditioning responses were obtained for penthyl acetate (PA) and ethyl lactate (EL) (S1 and S2, respectively); later flies were aversively trained to PA (S1) and 5 min after, post-conditioning responses were recorded. (**B, C**) S1 (PA) responses were completely depressed 5 min after training in both control flies and flies expressing *Rac1^{N17}*. Non-parametric Wilcoxon-paired test $p \leq 0.0078$; $n = 8–11$. No significant changes were detected to S2 (EL) for control animals. Non-parametric Wilcoxon-paired test $p = 0.5771$; $n = 11$. Nevertheless, a significant inhibition of post-trained response to S2 (EL) was observed in flies expressing *Rac1^{N17}*. Non-parametric Wilcoxon-paired test $p = 0.0391$; $n = 8$. Boxplots represent distribution of $\%\Delta F/F_o$ responses across the 5 s of odor presentation. The thick black bar below each trace represents the time of odor presentation. The asterisk's number shows N digits after the decimal.

The online version of this article includes the following source data and figure supplement(s) for figure 2:

**Source data 1.** Rac1 inhibition induces depression to sensory preconditioned odor S2 in MBON-γ1pedc>α/β using a different odor pair with orthogonal representation.

**Figure supplement 1.** *Rac1* inhibition does not induce depression to repeated odor presentation in MBON-γ1pedc>α/β when the electric shock is excluded.

**Figure supplement 1—source data 1.** Rac1 inhibition does not induce depression to repeated odor presentation in MBON-γ1pedc>α/β when the electric shock is excluded.

---

a T-maze by presenting animals with either the S1 (shock-paired odor) and a novel odor (NO), or the S2 (non-shocked, pre-paired odor) and an NO. As in canonical aversive olfactory conditioning experiments, each behavioral experiment was conducted using the reciprocal odor as S1. The final performance index (PI) was calculated by averaging the PI for each odor used as S1. Results were compared to flies trained with backwards conditioning. Despite observing evidence of sensory preconditioning by functional imaging using a similar protocol, we could not observe a behavioral memory to the non-shocked pre-paired odor (S2) (*Figure 4—figure supplement 1B*). At a behavioral level, sensory preconditioning has previously been observed in animals after repeated presentation of a pair of sensory cues. Thus, we preconditioned flies with 10 repeated presentations of odor pairings (S1/S2) before training (*Figure 4A*). Using an ISI of 1 s, this resulted in a significant behavioral aversive memory expression to the S2 odor in control flies (*Figure 4B*), consistent with our predictions based

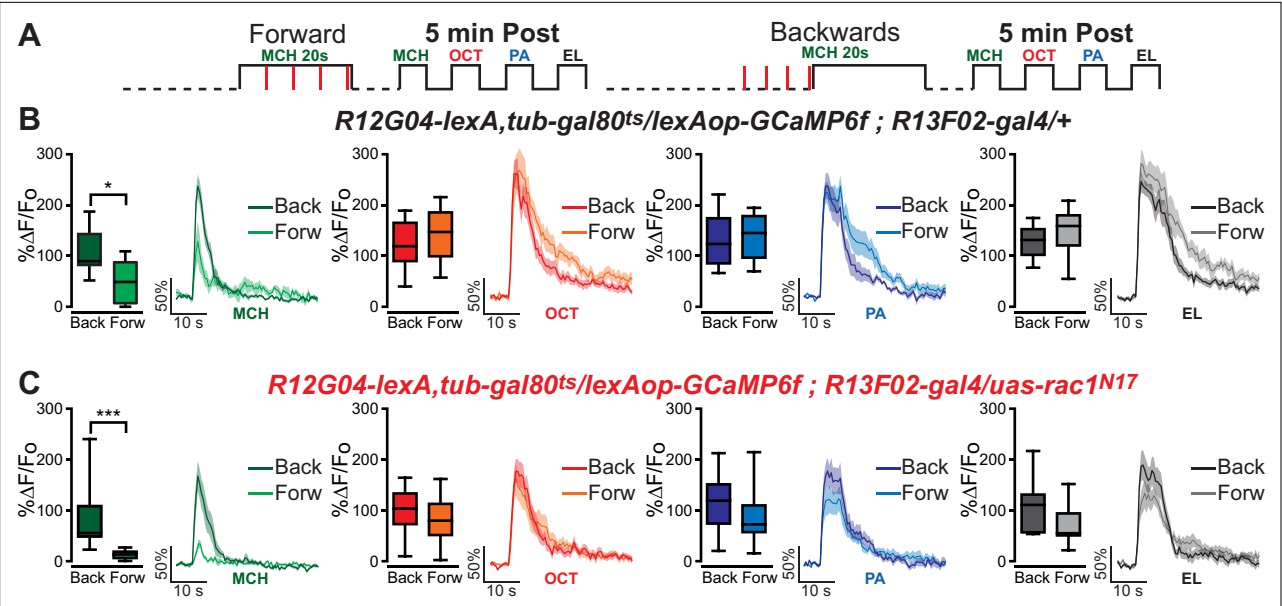

**Figure 3.** Excluding preconditioning eliminates depression to the S2 odor. (**A**) Diagram of experimental procedure (odor schedule): Flies were aversively trained to 4-methylcyclohexanol (MCH) (S1) and 5 min later post-conditioning responses to four odors were recorded (preconditioning was excluded). Responses were compared to flies trained using backwards conditioning. (**B, C**) S1 (MCH) responses were completely depressed 5 min after training in both control and flies expressing *Rac1^N17^*. Non-parametric Mann–Whitney test p ≤ 0.0147; n = 10–11. Neither 3-octanol (OCT), penthyl acetate (PA), nor ethyl lactate (EL) responses showed any significant reduction for both control and experimental flies. Non-parametric Mann–Whitney test p ≥ 0.072, n = 10–11. Boxplots represent distribution of %ΔF/F_o responses across the 5 s of odor presentation. The asterisk's number shows N digits after the decimal.

The online version of this article includes the following source data for figure 3:

**Source data 1.** Excluding preconditioning eliminates depression to the S2 odor.

on the sensory preconditioning physiological data. Furthermore, control flies did not demonstrate an aversive memory to the S2 odor when an ISI of 30 s was used (*Figure 4B*), as would be predicted from our physiological imaging data. These results also suggest that while a single odor pairing is sufficient to induce sensory preconditioned-related plasticity in the MBON-γ1pedc>α/β compartment, it is not enough for the S2 odor to drive the learned behavior. We suggest that the presentation of additional odor pairings recruits and induces sensory preconditioning-related plasticity in additional MB compartments, and it is the additive effect of multiple compartments that is necessary to drive the behavioral response. Similar phenomena have been observed before for classical conditioning where a short pairing of odor and shock – 1-s odor paired with 4 DAN photostimulation pulses – is enough to induce plasticity in MBON-γ1pedc>α/β compartment but not in MBON-α2sc. Longer training protocols – 1-min odor presentation paired with 120 DAN photostimulation pulses – induce plasticity in MBON-α2sc (*Hige et al., 2015*). Physiologically, preconditioning flies with MCH/OCT pairs repeated 10 times before training resulted in a robust depression in MBON-γ1pedc>α/β to the non-shocked pre-paired odor (CS2) in control flies (*Figure 4—figure supplement 2A*). These results were similar when PA/EL were used as the preconditioned odors (*Figure 4—figure supplement 2B*).

We next tested flies expressing *Rac1^N17^* in KC for evidence of behavioral sensory preconditioning when using 30-s and 5-min ISI between S1 and S2. As would be predicted from our physiological data, while a 30-s ISI resulted in no aversive memory to S2 in control flies, a significant aversive memory was observed in flies expressing *Rac1^N17^* (*Figure 4B*). This is consistent with our calcium imaging results in *Figure 1B*, in which a 30-s ISI results in a depression to S2 in *Rac1^N17^* flies but not in control flies. Further lengthening the ISI to 5 min eliminated behavioral sensory preconditioning both in controls and in flies expressing *Rac1^N17^* (*Figure 4B*), again mirroring the physiological results in the MBON-γ1pedc>α/β neuron (*Figure 1B*). Flies kept at 18°C to prevent expression of *Rac1^N17^* behaved as control flies and did not demonstrate behavioral evidence of sensory preconditioning when using an ISI of 30 min (*Figure 4—figure supplement 3*). Finally, we tested if *Rac1^N17^* expression in KC altered olfactory responses in the MB lobes. Interestingly, we could not find any change in the

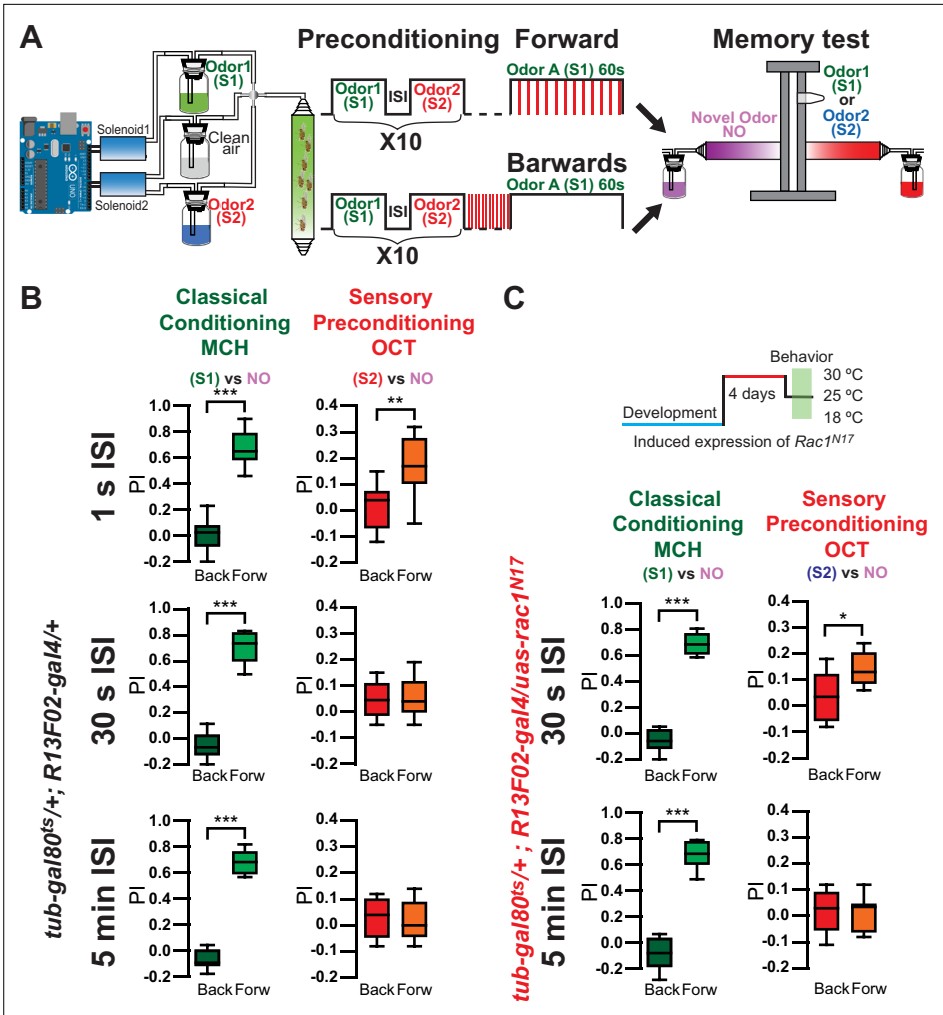

**Figure 4.** Repeated presentations of paired odors (S1/S2) induce behavioral sensory preconditioning. (**A**) Flies were trained using an Arduino microcontroller for precise odor delivery: Flies were preconditioned by 10 repeated presentations of S1/S2 odor pairs with 1-s interstimulus interval (ISI). Later flies were aversively trained to 4-methylcyclohexanol (MCH) (S1) by pairing 1-min odor presentation along 12 90 V, 1.25 s shocks. Right after training memory was tested in a T-maze by presenting either S1 vs. a novel odor (NO) or S2 vs. an NO. Performance index (PI) was compared to flies trained using backwards conditioning. (**B**) Memory to S1 was significantly different from flies trained with backwards conditioning, after 10× preconditioning presentations using 1-s, 30-s, or 5-min ISI. Non-parametric Mann–Whitney test p < 0.0001, *n* = 8–12. In contrast, memory to S2 was significantly different from flies trained with backwards conditioning, with 10× preconditioning presentations only using 1-s ISI in control flies. Non-parametric Mann–Whitney test p = 0.0019, *n* = 12. Memory to S2 was not significantly different from backwards conditioning, with 10× preconditioning presentations using 30-s or 5-min ISI in control flies. Non-parametric Mann–Whitney test p > 0.999, *n* = 8. (**C**) Memory to S1 in flies expressing *Rac1^N17^* was significantly different from flies trained with backwards conditioning, after 10× preconditioning presentations using 30-s or 5-min ISI. Non-parametric Mann–Whitney test p = 0.0002, *n* = 8. Memory to S2 in flies expressing *Rac1^N17^* was significantly different from flies trained with backwards conditioning, with 10× preconditioning presentations when using a 30-s ISI. Non-parametric Mann–Whitney test p = 0.0191, *n* = 8. Memory in flies expressing *Rac1^N17^* to S2 was not significantly different from backwards conditioning, with 10× preconditioning presentations when using 5-min ISI. Non-parametric Mann–Whitney test p = 0.8981, *n* = 8. The asterisk's number shows N digits after the decimal. The following source data and figure supplements are available for *Figure 4*.

The online version of this article includes the following source data and figure supplement(s) for figure 4:

**Source data 1.** Repeated presentations of paired odors (S1/S2) induce behavioral sensory preconditioning.

**Figure supplement 1.** A single presentation of paired odors (S1/S2) does not induce behavioral sensory preconditioning.

*Figure 4 continued on next page*

*Figure 4 continued*

**Figure supplement 1—source data 1.** A single presentation of paired odors (S1/S2) does not induce behavioral sensory preconditioning.

**Figure supplement 2.** Repeated presentations of paired odors (S1/S2) induces sensory preconditioning in MBON-γ1pedc>α/β.

**Figure supplement 2—source data 1.** Repeated presentations of paired odors (S1/S2) induces sensory preconditioning in MBON-γ1pedc>α/β.

**Figure supplement 3.** TARGET temperature control for *Rac1* inhibition does not show behavioral expression of sensory preconditioning.

**Figure supplement 3—source data 1.** TARGET temperature control for Rac1 inhibition does not show behavioral expression of sensory preconditioning.

**Figure supplement 4.** *Rac1* inhibition does not changes dynamics of calcium odor responses in γ lobes of Kenyon cells (KC).

**Figure supplement 4—source data 1.** Rac1 inhibition does not changes dynamics of calcium odor responses in γ lobes of Kenyon cells (KC).

---

calcium response dynamics in KC γ lobes, even when recordings were performed at a higher speed (28 Hz) (*Figure 4—figure supplement 4*).

Taken together, these results indicate that flies can undergo unimodal sensory preconditioning and that the small G protein *Rac1*, previously implicated in memory forgetting, gates the windows for this S1–S2 association.

## Ideas and speculations

Our results demonstrate that flies can infer value to non-reinforced odors based on the previous associative structure between odors. Memories are not discrete bits of events and associations that are added and accumulated in our brains. Rather, memories are dynamic; they undergo modifications and updates and are integrated into a coherent story that continuously incorporates new information to predict upcoming events better and to inform decision making. Within this complex associative structure, associations between seemingly valueless stimuli are fundamental. Here we present evidence that even the simple brain of a fly can form these associations.

How do these associations occur? Are S1–S2 associations the result of forming new or strengthening connections between KC following a Hebbian-like delay of synaptic reinforcement? If this is true, one could suggest a role for dopamine in this reinforcement. We know that DAN are essential for many types of reinforcement and even for non-associative learning like novelty detection (*Hattori et al., 2017*). Furthermore, we know dopamine neurons receive substantial feedback from both KC and MBON. Alternatively, S1–S2 associations could result from a shift in the stimulus representation during sensory preconditioning, probably by mechanisms similar to the hypothesis of excitability-dependent allocation of engram cells (or, in this case, neuronal representation) (*Yiu et al., 2014*; *Josselyn and Frankland, 2018*; *Josselyn and Tonegawa, 2020*).

Another interesting finding is that while a single odor pairing is sufficient to induce sensory preconditioned-related plasticity in the MBON-γ1pedc>α/β compartment, it is not enough for the S2 odor to drive the learned behavior. As mentioned above, we suggest that the presentation of additional odor pairings recruits and induces sensory preconditioning-related plasticity in additional MB compartments, and it is the additive effect of multiple compartments that is necessary to drive the behavioral response. Follow-up studies should include looking for the rules of sensory preconditioning-related plasticity in additional MB compartments. This finding has important implications for how memory systems work. There is accumulating evidence that memorable experiences are encoded in parallel, relatively independent memory traces, and it is the integration of these different memory traces that drives the most adequate behavior. Examples include the additive effect of aversive memory traces (*Aso et al., 2012*); the additive effect of contextual memory to aversive olfactory memories (*Zhao et al., 2019*); or the integration of parallel opposite memories during memory extinction (*Felsenberg et al., 2017*; *Felsenberg et al., 2018*).

Our data also demonstrate that inhibition of *Rac1* by expressing its dominant-negative form in the KC lengthens the time for an olfactory 'sensory buffer', allowing the linking of odors presented in

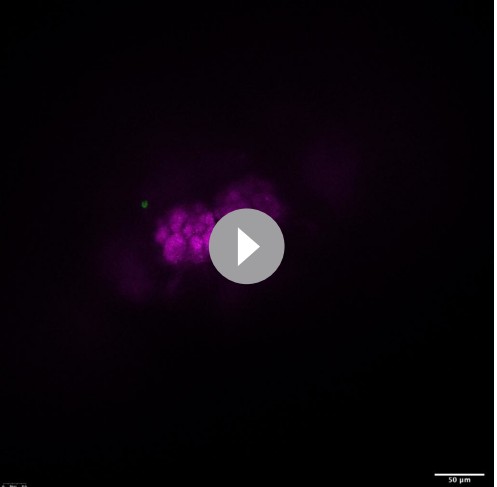

**Video 1.** Expression patterns of MB112C-splitGal4 driver expressing GFP. Brains where dissected and stained using anti-GFP and anti-nc82 as a counterstain. https://elifesciences.org/articles/79107/figures#video1

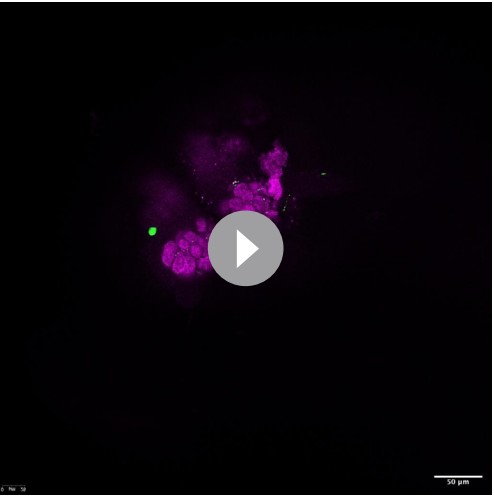

**Video 2.** Expression patterns of R12G04-LexA driver expressing GFP. Brains where dissected and stained using anti-GFP and anti-nc82 as a counterstain. https://elifesciences.org/articles/79107/figures#video2

sequence even when separated by at least 30 s. Interestingly, calcium responses in KC are not altered by the inhibition of *Rac1* (*Figure 4—figure supplement 4*). While calcium is a good proxy of neuronal activity within the KC, it does not necessarily directly translate to a readout of the neurotransmission output of the KC and its communication with downstream neurons.

A challenging question that remains for future study is on the nature of the circuit connections that link S1 and S2 during the sensory preconditioning phase. Once this S1–S2 link is formed, the key process by which we experimentally detect that link is as an aversive response to S2 during the testing phase. Does this aversive response to S2 result during the associative learning phase, in which the S1 presentation summons an S2 representation during CS/US pairing, and therefore, S2 forms a direct association with the punishment? Conversely, is the aversive response to S2 dependent on the memory retrieval phase, during which exposing flies to the S2 odor somehow recalls the S1-punishment association to influence the aversive response? This second possibility would imply that S2 is not

directly associated with punishment. Deciphering the neural circuit rules that support unimodal sensory preconditioning in the relatively simple fly brain will help us to unravel how other animals, including humans, use our nervous systems to build models of the world.

## Materials and methods
### *Drosophila* husbandry

Flies were cultured on standard medium at room temperature. Crosses, unless otherwise stated, were kept at 25°C and 70% relative humidity with a 12-hr light–dark cycle. The drivers used in this study include *MB112C-splitgal4* (*Aso et al., 2014*) (RRID: BDSC_68263), *R12G04-lexA* (*Jenett et al., 2012*) (RRID: BDSC_52448), and *R13F02-gal4* (*Jenett et al., 2012*) (RRID: BDSC_48571). Driver expression was verified by immunohistochemistry (*Videos 1–3*). Additional

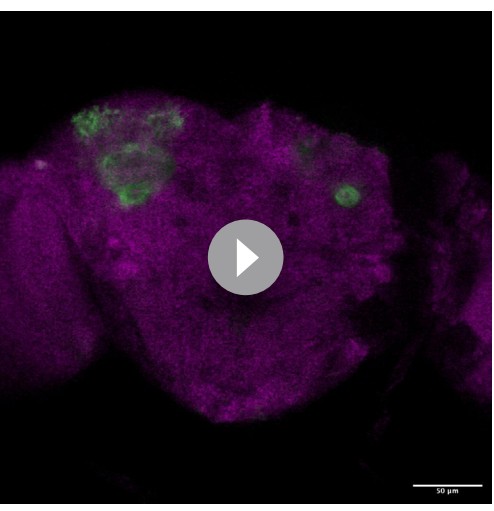

**Video 3.** Expression patterns of R13F02-Gal4 driver expressing GFP. Brains where dissected and stained using anti-GFP and anti-nc82 as a counterstain. https://elifesciences.org/articles/79107/figures#video3

transgene stocks included *uas-GCaMP6f* (*Chen et al., 2013*) (RRID: BDSC_42747), *lexAop-GCaMP6f* (*Chen et al., 2013*) (RRID: BDSC_44277), *uas-rac1$^{N17}$* (*Luo et al., 1994*) (RRID: BDSC_6292), and *tub-gal80$^{ts}$* (*McGuire et al., 2003*) (RRID: BDSC_7019).

To inhibit Rac1 activity in KC, the dominant-negative Rac1 (Rac1$^{N17}$) was expressed in adult flies using the Target system. Briefly, to restrict expression of transgenes to adult animals, fly crosses were kept at 18°C during development. After eclosion, 1- to 2-day-old flies were transferred to 30°C for 3–4 days for the induction of expression. Flies were then transferred to 25°C 1 hr before imaging or behavior. The Target system was used because inhibiting Rac1 in KC during development results in lethality. Control flies were subjected to exactly the same protocol but they did not contain the UAS transgene. Additional controls were performed by keeping the crosses at 18°C during development and after eclosion; these flies were then transferred to 25°C 1 hr before imaging.

### In vivo calcium imaging

For measuring calcium responses following conditioning, odor or shock delivery, we processed flies as previously described with some modifications (*Cervantes-Sandoval et al., 2017*; *Cervantes-Sandoval et al., 2020*). Briefly, a single fly was gently aspirated without anesthesia into a metal pipette to immobilize the head using proboscis aspiration. Once the head is immobilized, using a micromanipulator, the fly was inserted in a narrow slot the width of their body in a custom-designed recording chamber. The head was then fixed by gluing the eyes to the chamber using melted myristic acid. After this the fly was released from the metal pipette and the proboscis fixed with myristic acid to avoid brain movement during proboscis extension. Using a syringe needle, a small, square section of dorsal cuticle was removed from the head to allow optical access to the brain. Fresh saline (103 mM NaCl, 3 mM KCl, 5 mM HEPES (4-(2-hydroxyethyl)-1-piperazineethanesulfonic acid), 1.5 mM CaCl$_2$, MgCl$_2$, 26 mM NaHCO$_3$, 1 mM NaH$_2$PO$_4$, 10 mM trehalose, 7 mM sucrose, and 10 mM glucose [pH 7.2]) was perfused immediately across the brain to prevent desiccation and ensure the health of the fly. Then the fat bodies and trachea above the brain was removed. Using a 20× water-immersion objective and a Leica TCS SP8 II confocal microscope with a 488 nm argon laser, we imaged the MBON-γ1pedc>α/β neuron for 2 min at 2 Hz, during which stimuli was delivered starting at 30 s after imaging initiation. We used one HyD channel (510–550 nm) to detect *GCaMP6f* fluorescence. In order to detect small changes in the dynamics of odor responses in KC the recording speed was increased to 28 Hz.

### Odor and shock presentation

To deliver odors to flies under the microscope, a stream of air (500 ml/min) was diverted (via solenoids) from flowing through a clean 20 ml glass vial to instead flow through a 20-ml glass vial containing a 0.5 µl drop of pure odorant. This air stream was then serially diluted into a larger air stream (1500 ml/min) before traveling through Teflon tubing (~2.5 mm diameter) to reach the fly. Odors presented to the fly were continuously clear by a funnel connected to high vacuum pressure positioned right behind the recording chamber. To deliver shocks to flies under the microscope, a custom shock platform was made from shock grids used in the standard olfactory memory assays that consist of alternating +/− charged copper strips attached to an epoxy sheet. To simulate shock exposure given during the standard olfactory memory assay, the surface of the shock platform was positioned so that all six legs are touching but the fly could temporarily break contact by moving its legs. Both solenoids that control odor delivery and the Grass stimulator that delivers shocks were controlled by an Arduino microcontroller (Arduino Uno) with custom-made programs.

### Training under microscope programs

The regular training protocol followed for most experiments in the paper consisted of flies that were presented to preconditioning odors with 5 s of the first odor followed by variable ISI lengths (indicated in figures) during which clean air was presented, followed by 5 s of a second odor (non-associated odor) pre-training (MCH and OCT or PA and EL). The order of odor presentation was kept the same for all experiments and is indicated in each figure. Five minutes after these preconditioning recordings, flies were trained under the microscope by simultaneous presentation of a single 20-s odor pulse and four 90-V, 1.25-s electric shocks (5-s intershock interval). Five minutes after training, post-conditioning odor responses were recorded similarly to pre-responses. For control purposes, flies were trained with backwards training in which electric pulses were presented right before the onset of odor delivery.

## Behavior

Two- to five-day-old flies were used for all behavior experiments. Standard aversive olfactory conditioning experiments were performed as described (*Beck et al., 2000*) with some modifications. Precise control of odor delivery was achieved automatically using an Arduino microcontroller with custom-made programs to control a pair of solenoids. Briefly, a group of ~60 flies were loaded into a training tube where they received a preconditioned stimuli as indicated in each experiment. After preconditioning, flies were trained by 1 min of an odor paired with 12 pulses of 90-V electric shock (S1). We used OCT, MCH, PA, and EL as standard odorants. To measure memory, we transferred the flies into a T-maze where they were allowed 2 min to choose between two odors, either S1 and an NO or S2 and an NO.

## Immunostaining

Whole brains were isolated and processed with minor modifications from those described (*Jenett et al., 2012*). Brains were first incubated with primary antibodies including: rabbit polyclonal anti-GFP (1:1,000, Life Technologies Cat. # A11122, RRID: AB_10073917) and mouse monoclonal anti-nc82 (1:50, University of Iowa, DSHB, RRID: AB_2314866). Secondary antibodies included: anti-rabbit IgG conjugated to Alexa Fluor 488 (1:800, Life Technologies Cat. # A11008, RRID: AB_143165) and anti-mouse IgG conjugated to Alexa Fluor 633 (1:1000, Life Technologies Cat. # A21052, RRID: AB_141459). Images were collected using a 20× objective with a Leica TCS SP8 confocal microscope with 488 and 633 nm laser excitation.

## Quantification and statistical analysis

Fluorescence was acquired from a region of interest (ROI) drawn around the axon tract of MBON-γ1ped-c>α/β. Baseline was calculated using a Matlab program as the mean fluorescence across the 5 s before each odor presentation. This baseline was then used to calculate %$\Delta F/F_o$ for the complete recording. Bar graphs represent distribution of %$\Delta F/F_o$ responses across the 5 s of odor presentation. Solid lines in fluorescence traces represent mean %$\Delta F/F_o$ ± standard error (SE) (shaded area) across the odor responses.

Sample size was estimated using G*Power3.1 using preliminary data. All replicates in the manuscript were biological replicates. Statistics were performed using Prism 5 (GraphPad). All tests were two tailed and significance levels were set at $\alpha = 0.05$. The figure legends present the p values and comparisons made for each experiment. Unless otherwise stated, non-parametric tests were used for all imaging data.

## Acknowledgements

This work was supported by grants R21 MH117485-01A1 from the NIMH to IC-S, GU pilot research grant award ID162838 to IC-S, and BBRF Young Investigator Grant 30442. In addition, PS is supported NIA T32 AG071745 award.

## Additional information

### Funding

| Funder | Grant reference number | Author |
| --- | --- | --- |
| National Institute of Mental Health | R21MH117485-01A1 | Isaac Cervantes-Sandoval |
| Brain and Behavior Research Foundation | 30442 | Isaac Cervantes-Sandoval |
| National Institute on Aging | T32AG071745 | Prachi Shah |
| Georgetown University | ID162838 | Isaac Cervantes-Sandoval |

The funders had no role in study design, data collection, and interpretation, or the decision to submit the work for publication.

## Author contributions
Juan Martinez-Cervantes, Data curation, Investigation; Prachi Shah, Data curation, Formal analysis, Investigation; Anna Phan, Data curation, Investigation, Writing – original draft, Writing – review and editing; Isaac Cervantes-Sandoval, Conceptualization, Resources, Data curation, Formal analysis, Supervision, Funding acquisition, Investigation, Visualization, Methodology, Writing – original draft, Project administration, Writing – review and editing

## Author ORCIDs
Isaac Cervantes-Sandoval http://orcid.org/0000-0002-6372-7288

## Decision letter and Author response
Decision letter https://doi.org/10.7554/eLife.79107.sa1
Author response https://doi.org/10.7554/eLife.79107.sa2

## Additional files

### Supplementary files
• Transparent reporting form

### Data availability
All data generated or analyzed during this study are included in the manuscript and supporting files.

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
