## [Editor Report]

This paper shows that *Drosophila* can perform olfactory unimodal sensory preconditioning, an example of higher-order conditioning that may guide behaviour through inferred value. This is of conceptual significance for the brain, behavioural, and to some extent, the social sciences, because it shows that a conditioned response to a stimulus can occur even when the stimulus itself was never paired with punishment, for example.

---

## [Decision Letter]

**Decision letter after peer review:**

Thank you for submitting your article "Olfactory unimodal sensory preconditioning in *Drosophila*" for consideration by *eLife*. Your article has been reviewed by 3 peer reviewers, and the evaluation has been overseen by a Reviewing Editor and K VijayRaghavan as the Senior Editor. The reviewers have opted to remain anonymous.

Essential revisions:

These results are both interesting and important to the field. We have a few suggestions that we think will help this bring out and strengthen the claims made.

1. The writing: In the current form, the writing makes the story accessible largely to a specialist audience. That flies can learn through inferred value will be of interest to a wider audience and so, we're suggesting that the authors revisit their writing to make the data more accessible to the reader. Specifically, we're requesting that they revisit the introduction and abstract to better equip the reader with the specifics of the paper. This would include a more accessible description of sensory preconditioning (SPC) and the paradigms that model it; its prior demonstration in other insects; and the MBONs. We're also recommending that the authors reorganise their text and figures according to the convention (matched order in text and figures), and fix the numerous typos. Finally, we're requesting them to expand on their methodology.

2. Reorganisation of the story: We felt that a reorganisation of the data would greatly improve readability and make for a more convincing story. Our main suggestion is to focus on presenting all the data necessary to demonstrate that SPC is indeed occurring together. Specific suggestions on how to do this are in the detailed reviews below.

3. Discrepancy between behavioural and physiological responses: We were unclear as to why the authors chose the novel odours and why they did not use wild-type flies (CS) for this. So, we're requesting that the authors present their rationale for using novel odours and for using before-after instead of between-group comparisons. We also recommend that they use CS to address some of the concerns that have been raised regarding the controls.

*Reviewer #1 (Recommendations for the authors):*

Please check the whole manuscript for typos. There are also many wrong labels in the figures.

It was a bit confusing that the authors start the result section with figure2B. Either the authors provide a better description for this unusual beginning or change the figure arrangement.

Also, the Materials and methods is quite short. I was wondering if the odors in the test were always presented in the same order. S1, then S2, then other odors? Is this order necessary to induce the S2 effect? Especially in the 30s gap experiment, flies might not show the memory if the authors would be present in the "wrong order" compared to preconditioning.

To show that Rac1 inhibition might affect the duration of odor representation, the authors might have performed functional imaging in KCs if feasible.

Another question that came up is why the authors use a novel odor as a control after conditioning in behavioral experiments. Maybe that is confusing for the animals? Might they perform better if the S1 and S2 would be tested against the solvent or an empty tube? This also might explain discrepancies between imaging data and behavioral data.

*Reviewer #2 (Recommendations for the authors):*

In this manuscript, the authors provide physiological and behavioral evidence to support that fruit flies can perform olfactory unimodal sensory preconditioning, an example of higher-order conditioning. Flies were first exposed to a pair of odors (S1 and S2) with a 1-s inter-stimulus interval (ISI) before conditioning. Then odor S1 was given a negative value through aversive olfactory conditioning. After conditioning, the non-reinforced odor S2 was also found to be linked to the negative value. When the ISI was changed from 1 s to 30 s, flies failed to link odor S2 with the negative value, unless Rac1 activity was inhibited. Such sensory preconditioning can be observed in an aversive memory trace in a pair of MBON-γ1pedc>α/β neurons as well as behavioral performance. These results indicate that a simple brain may have the ability to guide behavior through inferred value and Rac1 may have a pivotal role in such behavior. The findings of sensory preconditioning are interesting and important, however, the behavioral data should be further strengthened by adding more controls. And calcium imaging results only partially explain the behavioral data, more discussion should be provided. More detailed information about methods is required. The writing needs to be improved.

1) The writing of this manuscript needs to be greatly improved. The text does not match the figures in some places. There are many other errors throughout the text. It makes reading the manuscript very difficult and time-consuming.

a) In line 95, "and vivo calcium imaging" should be "and in vivo calcium imaging", and "pairing and odor" should be "pairing an odor".

b) In line 105, the result description is inconsistent with Figure 2B.

c) In line 124, Figure 2A should be Figure 1B.

d) In line 129, Figure 1B should be Figure 1C.

e) In line 148, there is no Figure 2C.

f) In Figure 3A, "Barwards" should be "Backwards".

g) The first time the abbreviation KC is mentioned, the full form should be provided.

2) Is sensory preconditioning stable and significant at the behavioral level? The behavioral performance of the sensory preconditioning is weak and demonstrated by comparing forward and backward conditioning groups. As a new behavioral paradigm, more control groups such as naïve, "odor only" and "shock only" groups should be tested using wild-type flies like Canton-S.

3) Is one-second ISI sufficient to clear odor S1 and switch to odor S2? Is it possible that flies indeed sense mixed odors of S1 and S2? Is preconditioning effect can also be observed when S1-S2 mixed odors are exposed before conditioning? More evidence should be provided. It may affect the interpretation of the results.

4) As the authors mentioned in lines 183-186, the preconditioning effect observed in physiological trace in MBON-γ1pedc>α/β neurons (Figures 1 and 2) is not sufficient to explain the behavioral preconditioning in Figures 3 and 4. This point should be discussed more in the "ideas and speculations" section.

*Reviewer #3 (Recommendations for the authors):*

A weakness is seen in the way the data and arguments are presented, because key data belonging together are not shown and analysed together (and are not shown in the same way), and because the final "Ideas and Speculations" section of the Results and Discussion section reads somewhat disjointed.

Another weakness is that a rationale should be presented under which circumstances before-after comparisons are used as arguments, and when between-group comparisons are used.

Last but not least, the reviewer believes a better job could be done to make the manuscript more accessible for a non-expert audience, for example regarding the definition of wild-type animals, a plain-language rationale for the use of the transgenes, the logic of the Gal80ts/ target system including whether expressing the Rac transgene entails a gain or loss of Rac function, and what, thus, the colour code for presenting the genotypes means in terms of Rac function.

I) My main recommendation is to present together what belongs together, focusing on the most important issue, namely whether the evidence for sensory pre-conditioning (SPC) is indeed convincing. I discuss this for the physiological data, but the same applies to the behavioural data as well.

I think you could present together with the data from Figure 1, Figure 2-S1 (in between what are now Figure 1B and C), and Figure 2-S2 (after what is now Figure 1C). This would allow you to make your points for all the three mentioned requirements for a demonstration of SPC in one panel – plus: the broader time window for SPC upon Rac inhibition (?!) would be very nicely visible.

To simplify, you could present all the traces of recordings as supplements throughout.

You could then present a separate figure with all (sic) pre-post comparisons for that panel; "all" meaning for MCH and for OCT, and for both the forward and the backward cases.

There are a number of additional and rather unfortunate arrangements of the figures that make it hard to follow, almost creating 'stroop effects'. In addition to what was mentioned above, examples include:

If you deal with Figure 2 before Figure 1 in the Results section, you should also present these figures in that order. By the way, it might be a good idea to label the experiment in Figure 2 as a 'Pilot experiment' and present it as a supplement. This is because it is inconclusive with respect to all the three requirements for demonstrating SPC mentioned above.

Be consistent in presenting eg the control genotype towards the top and the experimental genotype towards the bottom, as in Figure 1 (this is apparently swapped in Figure 2-S1 and Figure 2-S2, and rotated by 90 degrees in Figure 2).

In general, I find your arguments regarding the discrepancy in terms of how many trials of pre-conditioning are needed for SPC at the physiological and behavioural levels reasonable.

As mentioned before, the "Ideas and Speculations" section of the Results and Discussion section lacks clarity. I like the idea of the "novelty" activations of DANs by untrained odours as a signal to establish S1-S2 associations.

At the conceptual level, one of the main future questions, I think, would be whether in addition to the S1-S2 link being formed during the first training phase, the key process to experimentally detect that link as SPC takes place during the second training phase (i.e. S1 calls up an S2-representation during this second training phase, such that this S2-representation is associated with punishment; meaning conditioned responding is based on an S2-punishment association), or whether this happens at the moment of the final test (i.e. at the moment of testing S2 calls up S1, which in turn calls up the S1-punishment association; meaning there is no direct S2-punishment association). One could rephrase this as: does S1-S2 pattern completion take place during the second training phase, and/or at the moment of the final test?

II) Further recommendations

More clarity is needed about the time intervals between the two training phases, and between the second training phase and the final test.

In a number of instances, articles or verbs appear to be missing. Some language polishing is advised.

The Introduction does not inform the reader that there have been earlier demonstrations of SPC in insects/ invertebrates. This may be read as a claim of primacy, although the authors, quite appropriately, do not explicitly make such a claim. In other words, not-citing the relevant literature unwittingly leads to an inappropriate, implicit claim of primacy.

Better do not introduce new terminology on the run, such as "S-S associations" line 205.

Better remove the money example from the Introduction as it refers to second-order conditioning, distracting from SPC as the subject of the study.

Line 148 refers to Figure 2C, which does not appear to exist.

---

## [Author Response]

Essential revisions:Reviewer #1 (Recommendations for the authors):Please check the whole manuscript for typos. There are also many wrong labels in the figures.It was a bit confusing that the authors start the result section with figure2B. Either the authors provide a better description for this unusual beginning or change the figure arrangement.

We have rearranged all figures as suggested by Reviewer 1 and other Reviewers.

Also, the Materials and methods is quite short. I was wondering if the odors in the test were always presented in the same order. S1, then S2, then other odors? Is this order necessary to induce the S2 effect? Especially in the 30s gap experiment, flies might not show the memory if the authors would be present in the "wrong order" compared to preconditioning.

We have expanded the methods details. Yes, odors were presented in the same order every time as indicated in each figure.

To show that Rac1 inhibition might affect the duration of odor representation, the authors might have performed functional imaging in KCs if feasible.

We have performed this experiment, as suggested by the reviewer. We expressed Gcamp6f in KC using R13F02-lexA>lexAop-GCaMP6f in flies expressing or not Rac1N17 using R13F02-gal4 and tub-gal80^ts^. Results did not show significant difference between these groups.

Another question that came up is why the authors use a novel odor as a control after conditioning in behavioral experiments. Maybe that is confusing for the animals? Might they perform better if the S1 and S2 would be tested against the solvent or an empty tube? This also might explain discrepancies between imaging data and behavioral data.

When we started the behavioral experiments, we used the solvent as the Reviewers suggested. We did not observe sensory preconditioning when a single odor pair was presented during the preconditioning phase. We later thought that using a third, well-balanced, novel odor during the test would give us a better chance to observe mild changes in the PI. The reasoning was that when using solvent, the baseline of S2 odor avoidance starts high. Therefore, small changes in this odor avoidance would be harder to see, in other words there is a smaller dynamic range. We reasoned that using an initially well-balanced novel odor would move the assay's baseline to zero, increasing the dynamic range. As shown in the results, a single odor pair pre-presentation was also unsuccessful when using a novel odor. After this, we decided to change two parameters: (1) the number of repeated pair presentations was increased to ten, and (2) we started constructing an odor delivery system using an Arduino to precisely control the timing of odor delivery. This new approach resulted in evidence of sensory preconditioning only when the preconditioning was repeated ten times. After this, we did not go back and test the solvent.

Reviewer #2 (Recommendations for the authors):In this manuscript, the authors provide physiological and behavioral evidence to support that fruit flies can perform olfactory unimodal sensory preconditioning, an example of higher-order conditioning. Flies were first exposed to a pair of odors (S1 and S2) with a 1-s inter-stimulus interval (ISI) before conditioning. Then odor S1 was given a negative value through aversive olfactory conditioning. After conditioning, the non-reinforced odor S2 was also found to be linked to the negative value. When the ISI was changed from 1 s to 30 s, flies failed to link odor S2 with the negative value, unless Rac1 activity was inhibited. Such sensory preconditioning can be observed in an aversive memory trace in a pair of MBON-γ1pedc>α/β neurons as well as behavioral performance. These results indicate that a simple brain may have the ability to guide behavior through inferred value and Rac1 may have a pivotal role in such behavior. The findings of sensory preconditioning are interesting and important, however, the behavioral data should be further strengthened by adding more controls. And calcium imaging results only partially explain the behavioral data, more discussion should be provided. More detailed information about methods is required. The writing needs to be improved.

As suggested by the Reviewer, all figures have been rearranged. Nevertheless, we have paid more attention to details to avoid any similar errors.

1) The writing of this manuscript needs to be greatly improved. The text does not match the figures in some places. There are many other errors throughout the text. It makes reading the manuscript very difficult and time-consuming.

We have corrected these mistakes.

a) In line 95, "and vivo calcium imaging" should be "and in vivo calcium imaging", and "pairing and odor" should be "pairing an odor".

This has been superseded.

b) In line 105, the result description is inconsistent with Figure 2B.

This has been superseded.

c) In line 124, Figure 2A should be Figure 1B.

This has been superseded.

d) In line 129, Figure 1B should be Figure 1C.

This has been superseded.

e) In line 148, there is no Figure 2C.

This has been superseded.

f) In Figure 3A, "Barwards" should be "Backwards".

This has been corrected.

g) The first time the abbreviation KC is mentioned, the full form should be provided.

This has been corrected.

2) Is sensory preconditioning stable and significant at the behavioral level? The behavioral performance of the sensory preconditioning is weak and demonstrated by comparing forward and backward conditioning groups. As a new behavioral paradigm, more control groups such as naïve, "odor only" and "shock only" groups should be tested using wild-type flies like Canton-S.

We used backwards conditioning because these control flies are exposed to all the same stimuli but in a different order. So it is more adequate than testing odors only and shock only. Furthermore, we continuously performed behavior with naïve animals to ensure we had a stable odor balance before experiments. Therefore naïve only control gives a behavioral π score of zero. Finally, during the imaging experiments, we did show that the presentation of odors only, backwards conditioning, and excluding the preconditioning phase all resulted in non-observable physiological sensory preconditioning.

3) Is one-second ISI sufficient to clear odor S1 and switch to odor S2? Is it possible that flies indeed sense mixed odors of S1 and S2? Is preconditioning effect can also be observed when S1-S2 mixed odors are exposed before conditioning? More evidence should be provided. It may affect the interpretation of the results.

The Reviewer makes a good point here. In the classical sensory preconditioning experiments, stimuli of two different modalities are usually presented together or with some overlap. Therefore, sensory preconditioning still occurs if stimuli are presented together. Because we are working with unimodal sensory preconditioning, we decided to avoid odor mixing to simplify possible interpretations.

During imaging, odors delivered to the flies are on a continuous flow of air. Therefore, the odors are pushed by clean air as soon as solenoids valves switch from odor back to clean air. In addition, odors are immediately cleared by a funnel connected to a high vacuum pressure right behind the recording chamber.

During behavior, because the system works with a closed continuous vacuum, odors are cleared immediately after being presented.

Finally, and more importantly, the Rac1 inhibition experiments show that sensory preconditioning can still occur even when odors are presented 30 s apart.

4) As the authors mentioned in lines 183-186, the preconditioning effect observed in physiological trace in MBON-γ1pedc>α/β neurons (Figures 1 and 2) is not sufficient to explain the behavioral preconditioning in Figures 3 and 4. This point should be discussed more in the "ideas and speculations" section.

We have included a discussion about this in “ideas and speculation” section.

Reviewer #3 (Recommendations for the authors):A weakness is seen in the way the data and arguments are presented, because key data belonging together are not shown and analysed together (and are not shown in the same way), and because the final "Ideas and Speculations" section of the Results and Discussion section reads somewhat disjointed.

We have rearranged the data as suggested by the Reviewer. We have added specific points to the ideas and speculations section. In particular the disconnect between the physiological and behavioral data

Another weakness is that a rationale should be presented under which circumstances before-after comparisons are used as arguments, and when between-group comparisons are used.

In the Material and Methods section, we have added the rationale for selecting the different ways of analyzing the groups—especially the before-after vs. forward-backward comparisons.

Last but not least, the reviewer believes a better job could be done to make the manuscript more accessible for a non-expert audience, for example regarding the definition of wild-type animals, a plain-language rationale for the use of the transgenes, the logic of the Gal80ts/ target system including whether expressing the Rac transgene entails a gain or loss of Rac function, and what, thus, the colour code for presenting the genotypes means in terms of Rac function.

We have edited the manuscript in an attempt to make it more accessible for non-expert audiences as suggested by Reviewer 3.

I) My main recommendation is to present together what belongs together, focusing on the most important issue, namely whether the evidence for sensory pre-conditioning (SPC) is indeed convincing. I discuss this for the physiological data, but the same applies to the behavioural data as well.I think you could present together with the data from Figure 1, Figure 2-S1 (in between what are now Figure 1B and C), and Figure 2-S2 (after what is now Figure 1C). This would allow you to make your points for all the three mentioned requirements for a demonstration of SPC in one panel – plus: the broader time window for SPC upon Rac inhibition (?!) would be very nicely visible.To simplify, you could present all the traces of recordings as supplements throughout.You could then present a separate figure with all (sic) pre-post comparisons for that panel; "all" meaning for MCH and for OCT, and for both the forward and the backward cases.

We have rearranged the data to be as close as possible to the Reviewer 3 suggestions.

There are a number of additional and rather unfortunate arrangements of the figures that make it hard to follow, almost creating 'stroop effects'. In addition to what was mentioned above, examples include:If you deal with Figure 2 before Figure 1 in the Results section, you should also present these figures in that order. By the way, it might be a good idea to label the experiment in Figure 2 as a 'Pilot experiment' and present it as a supplement. This is because it is inconclusive with respect to all the three requirements for demonstrating SPC mentioned above.Be consistent in presenting eg the control genotype towards the top and the experimental genotype towards the bottom, as in Figure 1 (this is apparently swapped in Figure 2-S1 and Figure 2-S2, and rotated by 90 degrees in Figure 2).In general, I find your arguments regarding the discrepancy in terms of how many trials of pre-conditioning are needed for SPC at the physiological and behavioural levels reasonable.As mentioned before, the "Ideas and Speculations" section of the Results and Discussion section lacks clarity. I like the idea of the "novelty" activations of DANs by untrained odours as a signal to establish S1-S2 associations.At the conceptual level, one of the main future questions, I think, would be whether in addition to the S1-S2 link being formed during the first training phase, the key process to experimentally detect that link as SPC takes place during the second training phase (i.e. S1 calls up an S2-representation during this second training phase, such that this S2-representation is associated with punishment; meaning conditioned responding is based on an S2-punishment association), or whether this happens at the moment of the final test (i.e. at the moment of testing S2 calls up S1, which in turn calls up the S1-punishment association; meaning there is no direct S2-punishment association). One could rephrase this as: does S1-S2 pattern completion take place during the second training phase, and/or at the moment of the final test?

We acknowledge these comments from the Reviewer 3. He makes very interesting points. We have added these to the discussion and speculations section.

II) Further recommendationsMore clarity is needed about the time intervals between the two training phases, and between the second training phase and the final test.

The time between the first and the second training phases is 5 minutes. The time between the second training phase and the test phase was 5 min. We made sure this is explicit in the material and methods.

In a number of instances, articles or verbs appear to be missing. Some language polishing is advised.

We have polished the English writing of the manuscript.

The Introduction does not inform the reader that there have been earlier demonstrations of SPC in insects/ invertebrates. This may be read as a claim of primacy, although the authors, quite appropriately, do not explicitly make such a claim. In other words, not-citing the relevant literature unwittingly leads to an inappropriate, implicit claim of primacy.

We have including previous reports of SPC in the new version of the manuscript.

Better do not introduce new terminology on the run, such as "S-S associations" line 205.Better remove the money example from the Introduction as it refers to second-order conditioning, distracting from SPC as the subject of the study.Line 148 refers to Figure 2C, which does not appear to exist.

We addressed these comments as suggested.